# Radiomics in Lung Metastases: A Systematic Review

**DOI:** 10.3390/jpm13020225

**Published:** 2023-01-27

**Authors:** Michela Gabelloni, Lorenzo Faggioni, Roberta Fusco, Igino Simonetti, Federica De Muzio, Giuliana Giacobbe, Alessandra Borgheresi, Federico Bruno, Diletta Cozzi, Francesca Grassi, Mariano Scaglione, Andrea Giovagnoni, Antonio Barile, Vittorio Miele, Nicoletta Gandolfo, Vincenza Granata

**Affiliations:** 1Nuclear Medicine Unit, Department of Translational Research, University of Pisa, 56126 Pisa, Italy; 2Academic Radiology, Department of Translational Research, University of Pisa, 56126 Pisa, Italy; 3Medical Oncology Division, Igea SpA, 80013 Naples, Italy; 4Division of Radiology, Istituto Nazionale Tumori IRCCS Fondazione Pascale—IRCCS di Napoli, 80131 Naples, Italy; 5Department of Medicine and Health Sciences V. Tiberio, University of Molise, 86100 Campobasso, Italy; 6General and Emergency Radiology Department, “Antonio Cardarelli” Hospital, 80131 Naples, Italy; 7Department of Radiology, University Hospital “Azienda Ospedaliera Universitaria delle Marche”, 60126 Ancona, Italy; 8Department of Clinical, Special and Dental Sciences, Università Politecnica delle Marche, 60121 Ancona, Italy; 9Department of Diagnostic Imaging, Area of Cardiovascular and Interventional Imaging, Abruzzo Health Unit 1, 67100 L’Aquila, Italy; 10Italian Society of Medical and Interventional Radiology, SIRM Foundation, 20122 Milan, Italy; 11Department of Emergency Radiology, Careggi University Hospital, 50134 Florence, Italy; 12Department of Precision Medicine, Università degli Studi della Campania “Luigi Vanvitelli”, 80138 Naples, Italy; 13Department of Surgery, Medicine and Pharmacy, University of Sassari, 07100 Sassari, Italy; 14Department of Biotechnological and Applied Clinical Sciences, University of L’Aquila, 67100 L’Aquila, Italy; 15Diagnostic Imaging Department, Villa Scassi Hospital-ASL 3, 16149 Genoa, Italy

**Keywords:** lung metastases, radiomics, oncologic imaging, computed tomography, positron emission tomography

## Abstract

Due to the rich vascularization and lymphatic drainage of the pulmonary tissue, lung metastases (LM) are not uncommon in patients with cancer. Radiomics is an active research field aimed at the extraction of quantitative data from diagnostic images, which can serve as useful imaging biomarkers for a more effective, personalized patient care. Our purpose is to illustrate the current applications, strengths and weaknesses of radiomics for lesion characterization, treatment planning and prognostic assessment in patients with LM, based on a systematic review of the literature.

## 1. Introduction

The lung is the second commonest site for metastases from extra-thoracic cancers, occurring in 20–54% of metastatic patients. Pulmonary involvement may result from direct tumor invasion or from hematogenous or lymphatic spread of tumor cells. In the adult population, the most frequent primary tumors that disseminate to the lung include breast, colorectal, renal, uterine cancer, leiomyosarcoma and head and neck carcinoma. In limited cases, the primary tumor origin cannot be identified.

Treatment options for patients with lung metastases (LM) include surgery, radiotherapy, and local or systemic therapies. In recent years, the efficacy of systemic therapies has improved due to advancements in treatment strategies and the introduction of molecular targeted drugs. In parallel, the increasing availability of new therapeutic approaches has created the need to identify beforehand which patients are eligible to a given specific treatment or another, raising the requirements for proper disease staging to a more evolved and complex level [1,2,3,4,5,6,7,8,9].

Differentiating pulmonary metastases from primary or benign lung lesions is of paramount importance. Among imaging modalities, multidetector computed tomography (MDCT) and 18-fluorodeoxyglucose positron emission tomography/computed tomography (^18^F-FDG PET/CT) play a major role in the detection, characterization and follow-up of lung cancer (either primary or LM). While MDCT is the mainstay for morphological assessment (allowing to evaluate features such as lesion size, morphology and growth rate), ^18^F-FDG PET/CT can complement morphological imaging by providing functional information related to tumor metabolism.

Despite advancements in MDCT and ^18^F-FDG PET/CT technologies, the diagnostic accuracy of both modalities can be limited by false positive (e.g., increased contrast enhancement or ^18^F-FDG uptake in non-neoplastic conditions, such as infections or inflammatory lesions) and false negative findings (as can happen, e.g., with poorly vascularized malignancies and/or those with relatively low ^18^F-FDG metabolism). Lung biopsy is the diagnostic gold standard to distinguish between primitive lung tumors or LM, but it is an invasive technique that may not be performed in all cases, cannot be repeated indefinitely, and may be burdened by more or less serious complications. Another inherent downside to conventional biopsy is the fact that, even if guided by imaging, it can provide information solely related to the specific tumor specimen from which cells are sampled, only allowing a limited assessment of tumor heterogeneity with no ability to interrogate the whole tumor structure [10,11,12,13,14].

Minimally invasive techniques for the quantitative evaluation of biomarkers [15,16,17,18,19,20,21,22,23] and response to therapies [24,25,26,27,28,29,30,31,32] have emerged, such as liquid biopsy [33,34,35,36,37,38]. Radiomics is an emerging methodology aiming to convert diagnostic images into information that reflect pathophysiological properties of the tumor [39]. By tapping the computational power of artificial intelligence (AI) systems, a large amount of information that go beyond visual image assessment can be collected through the extraction and analysis of radiomics features from medical images, obtained in a routine setting with conventional imaging protocols (e.g., for tumor staging and follow-up). This approach has the potential to enhance options to carry out patient-specific diagnostic and prognostic evaluations, allowing e.g., to predict early treatment response and avoid undue over- or undertreatment depending on the biological aggressiveness of each patient-specific disease, and paving the way to a more individualized patient management [22,40,41,42,43,44,45,46,47,48,49,50,51,52,53,54,55,56,57,58,59,60,61,62,63,64,65,66,67,68]. Other advantages of radiomics over conventional biopsy include the possibility to capture the tumor tissue in its entirety (providing information regarding its clonal heterogeneity and the tumor microenvironment along with its stromal component), to guide biopsy towards specific tumor areas, and to be repeatable at virtually any time during the disease course, allowing longitudinal monitoring with improved opportunities of optimizing treatment strategies [39,43,44,45].

Our aim is to explore the current literature regarding the potential of radiomics in the field of diagnosis, treatment planning and outcome prediction of LM.

## 2. Methods

We systematically searched the literature to assess the role of radiomics in the management of LM from diagnosis to prognosis. We searched the PubMed (https://pubmed.ncbi.nlm.nih.gov, assessed on 20 November 2022) and Scopus (https://www.scopus.com, assessed on 20 November 2022) databases using a combination of the following search terms: ((“radiomics” OR “machine learning”) AND (metastases OR metastasis) AND (“lung” OR “pulmonary”)). The search period ranged from January 2015 to November 2022, and the search was performed on 20 November 2022. Only articles written in English language were selected. The search was restricted to “article, abstract and keys” in the Scopus database.

To improve the quality of inclusion criteria, we followed the Preferred Reporting Items for Systematic Reviews and Meta-Analyses (PRISMA) guidelines [69]. The radiomics quality score (RQS) was calculated to assess the characteristics and the quality of the methodology of the studies taken into consideration, as described by Lambin et al. [70].

## 3. Results

All retrieved publications (*n* = 890) were separately uploaded to EndNote™, and all duplicate records were eliminated (*n* = 281).

Two reviewers (M.G., L.F.) manually screened all articles by title and abstract, and if eligible, their full text was retrieved. Exclusion criteria were as follows: (1) animal studies, (2) reviews, posters and conference papers, (3) entries related to predicting the onset of LM from primitive tumors, and (4) entries related to computer aided detection of LM.

544 studies were excluded due to having irrelevant titles and abstracts. Of the remaining articles, 43 were excluded due to being conference papers, posters and reviews, 2 due to being related to LM detection, 3 due to being animal studies, and 9 due to being related to predicting the onset of LM from primitive tumors (Figure 1).

Finally, 8 articles were collected and divided into the following categories: (1) distinguishing histological subtypes (*n* = 6), (2) evaluation of mutational status (*n* = 1) and (3) prognostic assessment (*n* = 1) (Table 1).

## 4. Discussion

Differentiating primary lung cancer from LM is of great clinical interest because of the profound differences in their prognostic and therapeutic implications [79,80]. Furthermore, a solitary pulmonary nodule may be more difficult to interpret in patients with a history of cancer, as it may be a primary lung tumor, a metastasis, or a benign lesion [81,82,83]. From an imaging viewpoint, the ability to recognize various diseases based on qualitative criteria relies on observer expertise and experience, whereas quantitative parameters, such as pulmonary nodule diameter and/or growth rate, are surrogate biomarkers of malignancy [84,85,86,87].

In this context, 3 articles evaluated the role of radiomics in differentiating primary lung tumors from metastases, of which 2 analyzed PET/CT images and 1 MDCT images. Kirienko et al. [73] evaluated 534 patients who underwent ^18^F-FDG PET/CT for characterization of lung nodule or for staging of a suspected lung tumor before biopsy, with the aim to evaluate the ability of radiomics to discriminate primitive lung tumor from LM and to predict the histological subtypes among primary lung cancers. They demonstrated that texture features obtained from PET images were able to differentiate between primary lung cancer and metastases (AUC > 0.90), whereas the analysis of co-registered CT data showed limited ability to distinguish between the two groups. Furthermore, the data obtained from PET images showed higher ability to discriminate between histological subtypes than those derived from CT images. More specifically, as to differentiation between primary subgroups based on CT features, the AUCs in the training and validation groups were 0.81 ± 0.02 and 0.69 ± 0.04 for adenocarcinoma versus squamous cell carcinoma or other histological subtypes, 0.85 ± 0.02 and 0.70 ± 0.05 for squamous cell carcinoma versus adenocarcinoma or other histological subtypes, and 0.77 ± 0.03 and 0.57 ± 0.05 for other histological subtypes versus squamous cell carcinoma or adenocarcinoma. The same analyses for the PET data revealed AUCs of 0.90 ± 0.10 and 0.80 ± 0.04, 0.80 ± 0.02 and 0.61 ± 0.06, and 0.97 ± 0.01 and 0.88 ± 0.04, respectively.

In a study performed on 769 patients, Zhou et al. [78] showed that ^18^F-FDG PET/CT features allowed to distinguish primary lung tumors from LM (AUC = 0.98), and that the results derived from the CT dataset were generally poorer than those obtained from the PET dataset. Furthermore, they achieved a good performance in discriminating lung adenocarcinoma from squamous cell carcinoma (AUC = 0.89) by using a combination of gradient boosting decision tree (GBDT) feature selection method with GBDT classification in the PET dataset. The combination of the GBDT feature selection method with the random forest (RF) classification had the highest AUC of 0.83 in the CT dataset. Of note, most of the decision tree (DT)-based models were overfitted, revealing that the classification method was inappropriate for usage in clinical practice.

In the aforementioned papers, the poorer performance obtained with CT data could be due to the fact that the CT dataset used for anatomical localization of PET/CT scans had a lower tissue contrast resolution (resulting from the lack of intravenous contrast material administration) and a lower overall quality than regular MDCT images used for radiological diagnosis, possibly introducing a bias in the determination of radiomics features. However, these findings could underscore the ability of texture analysis of PET images to detect the expression of underlying biological processes as revealed by an increased ^18^F-FDG uptake.

Zhong et al. [77] analyzed MDCT images of 97 s primary lung cancers and 155 LM, and constructed a nomogram model integrating clinical data, imaging characteristics (such as distribution of lesions, central or peripheral type, contours, and spiculation), and radiomics features. They achieved an excellent (0.94 and 0.90 in the training and validation cohorts, respectively) discriminative capability to distinguish LM from second primitive lung cancer in patients with a history of cancer. The radiomics model alone had good discriminative performance, with an AUC of 0.87 (95% confidence interval, 0.81–0.92) in the training set and 0.76 (95% confidence interval, 0.67–0.85) in the validation set. Moreover, the decision curve analysis (DCA) showed that the comprehensive model had a higher clinical value than that without the radiomics score.

At primary tumor staging or follow-up, patients frequently present with pulmonary nodules with indeterminate characteristics, such as <1 cm in diameter, single or double, no typical CT signs of malignancy, and no elevation of ^18^F-FDG standardized uptake value (SUV) at PET-CT imaging. These indeterminate lung nodules usually undergo long-term follow-up, bringing additional cost and patient anxiety and possibly delaying treatment. Therefore, a noninvasive diagnostic tool allowing a reliable and immediate characterization of such lesions would fulfil a highly unmet clinical need.

Hu et al. [72] evaluated 194 patients with colorectal cancer and MDCT finding of at least one indeterminate pulmonary nodule sized 5–20 mm in diameter. Three models were generated (i.e., a clinical model with significant clinical risk factors, a radiomics model with radiomics features constructed by the least absolute shrinkage and selection operator (LASSO) algorithm, and a clinical-radiomics model with significant variables selected by the stepwise logistic regression) to quantitatively assess the risk of such patients to develop LM. A nomogram was built based on the best performing model, and DCA was applied to test the clinical usefulness. The rad-score DCA showed more benefit than clinical DCA (based on N stage, chronicity, and size of the nodule) in predicting the risk of LM. Moreover, the clinical-radiomics nomogram was successfully developed with a favorable discrimination in the training cohort (AUC = 0.92, 95% confidence interval: 0.88–0.97) and the validation cohort (AUC = 0.92, 95% confidence interval: 0.85–0.98) and good calibration, and it achieved the greatest clinical usefulness, being capable of discriminating LM from non-metastatic nodules (84.9% sensitivity and 91.1% specificity in the training cohort). A higher nodal stage was found to be a significant predictor of metastases in patients with colorectal cancer and indeterminate pulmonary nodules (warranting a closer monitoring for early detection and follow-up of LM in this patient subset), and metachronous nodules were closely correlated with LM occurrence and had an excess predictive impact compared with N stage.

Liu et al. [74] analyzed radiomics features from contrast enhanced MDCT images to differentiate benign nodules from metastatic pulmonary nodules from colorectal cancer. A total of 320 nodules sized less than or 1 cm in diameter were evaluated, of which 200 metastatic nodules were included in the training cohort, 60 benign nodules in one verification cohort, and 60 metastatic nodules in another verification cohort. All nodules were divided into four groups according to their maximum diameter. Through cross-validation on 100 experiments, 11 features remained stable for more than 90 times in LM, but not in benign nodules. Such stability may be related to the essential characteristics of metastatic nodules, possibly representing a relevant factor to distinguish metastatic pulmonary nodules from benign ones. Of note, 8 of 11 features belong to ‘CoLIAGe’ (i.e., Co-occurrence of Local Anisotropic Gradient Orientations), a radiomics descriptor that identifies differences in the local entropy pattern and can differentiate subtle pathology differences from similar morphological manifestations, unlike morphological descriptors (such as shape and edges), which can be similar in small benign nodules and metastatic nodules.

Shang et al. [76] explored the role of radiomics to differentiate lung metastatic nodules from breast, colorectal and renal cancer by means of MDCT radiomics features. They performed a retrospective analysis including 252 LM from 78 patients, which were randomly divided into a training cohort (*n* = 176) and a test cohort (*n* = 76). The metastases originated from colorectal cancer (*n* = 97), breast cancer (*n* = 87), and renal carcinoma (*n* = 68), and additional 77 LM were used for external validation. A three-class model was built using the LASSO method, showing a good discriminative performance for the various tumor histotypes (AUC: colorectal cancer LM vs. renal carcinoma LM 0.84, breast cancer LM vs. colorectal cancer LM 0.80, breast cancer LM vs. renal carcinoma LM 0.94), along with AUCs of 0.77, 0.78, and 0.84 in the external validation cohort. Interestingly, breast cancer LM showed a low value of normalized run variance, maximum probability, and dependence nonuniformity. This may indicate more homogeneity in texture patterns, possibly due to dense fibrous tissue hyperplasia in breast cancer lesions, as compared with renal carcinoma (which is prone to intratumor hemorrhage and necrosis/cystic degeneration) and colorectal cancer (which tends to have fewer stroma elements inside it).

Treatment options for LM may include local therapies, such as thermal ablation techniques and stereotactic ablative body radiation [88,89,90,91,92,93,94,95,96], chemotherapy [97,98] and, in selected cases, surgery [99,100,101]. The introduction of new systemic treatments, including immunotherapy and target therapies, has improved the prognosis of patients with metastatic tumors, but only a subset of mutated patients can benefit from such treatments [102]. Angus et al. [71] attempted to detect radiomics features related to BRAF mutations in LM from melanoma in patients undergoing pre-treatment MDCT. 540 radiomics were extracted from 169 lung lesions detected in 103 patients (51 BRAF-mutant, 52 BRAF-wild type), and a combination of machine learning methods was used to build BRAF decision models based on radiomics features and Lung Image Database Consortium (LIDC) criteria. No radiomics features were found to be able to differentiate BRAF-mutant and BRAF-wild type LM, with models based on radiomics features and LIDC criteria performing as poorly as guessing. A potential explanation for this finding it that as the NRAS and BRAF genes are both involved in the MPAK pathway, activating mutations of the NRAS gene in BRAF-wild type melanoma or the BRAF gene in BRAF-mutant melanoma may lead to a similar phenotype.

To our knowledge, only Miao et al. [75] evaluated the role of radiomics to predict prognosis in patients with LM. They investigated radiomics features from contrast-enhanced MDCT examinations of 51 patients to predict the effectiveness of epirubicin combined with ifosfamide as first-line treatment in LM from soft tissue sarcoma. Lung metastases were used as target lesions (total *n* = 86), and patients were split into a progression group (*n* = 29), a stable group (*n* = 34), and a partial response group (*n* = 23). In total, 851 radiomics features were extracted for each target lesion, and then narrowed down to 2 radiomics features (wavelet-HHH_First Order Mean and wavelet-LHL_GLRLM Long Run Low Grey Level Emphasis) by dimensionality reduction. Such features were used to build a decision tree classifier model with a good predictive value (AUC = 0.91, 95% confidence interval: 0.858–0.969 in the training group, and AUC = 0.85, 95% confidence interval: 0.72–0.96 in the testing group). This finding has a relevant practical value because, if there is a high likelihood of disease progression according to the predictive model, alternative treatment strategies can promptly be enacted, potentially improving disease-free survival and overall quality of life.

Despite its potential in providing an added value over conventional imaging approaches in the management of patients with LM and other cancers, radiomics is still far from being universally adopted in clinical practice. This is due to several factors, including a lack of harmonization of imaging protocols, clinical validation issues, and an overall poor scientific quality of the studies in the field [43,45,103,104]. In line with this scenario, a RQS of 27.8% (range 22.2–38.9%) was calculated for the 8 articles retrieved in this review (Table 2), consistent with other reviews focused on radiomics [105,106,107,108,109,110,111,112,113,114,115,116,117,118,119,120].

Moreover, in all studies patient enrollment was retrospective in design, validation was performed on datasets from the same institutions, and data access was not granted to the public. Three studies (37.5%) [55,56,68] evaluated the potential applicability of radiomics models in a clinical setting by means of DCA. In most studies (75%) [71,72,74,75,76,77] multiple segmentations were performed to evaluate the robustness of radiomics features in relation to segmentation variability.

This review has some limitations. Firstly, we selected only studies in which radiomics features had been obtained from LM, leaving out any studies related to LM prediction from primitive tumors. Secondly, the high variability of the studies under investigation (e.g., regarding methodology) makes it difficult to compare data across them.

## 5. Conclusions

Radiomics is a relatively recent field of research in rapid evolution, and the potential use of radiomics for LM is a niche subfield, as demonstrated by the scarcity of studies collected in this review. Their lack of clinical relevance assessment and their retrospective design in a single center setting without external validation are shortcomings that currently curtail the robustness and clinical applicability of the study findings. Furthermore, no solid data have been collected so far demonstrating a correspondence between specific textural features and biological processes and explaining it in terms of radiomics-pathology correlations. In this setting, specific initiatives (such as the Image Biomarker Standardization Initiative, IBSI) have been undertaken to promote the adoption of a standardized approach for the definition, nomenclature, and calculation of radiomics features, resulting in improved statistical reliability as long as calculation settings are harmonized too. A greater standardization of radiomics methods and their validation on larger patient samples obtained from multicenter studies (possibly based on standardized image acquisition protocols), as well as a better integration of radiomics software in real world clinical and radiological environments [121,122,123,124,125,126,127], could be key to increase the role of radiomics in diagnosis, treatment planning and individual outcome prediction in patients with LM.

## Figures and Tables

**Figure 1 jpm-13-00225-f001:**
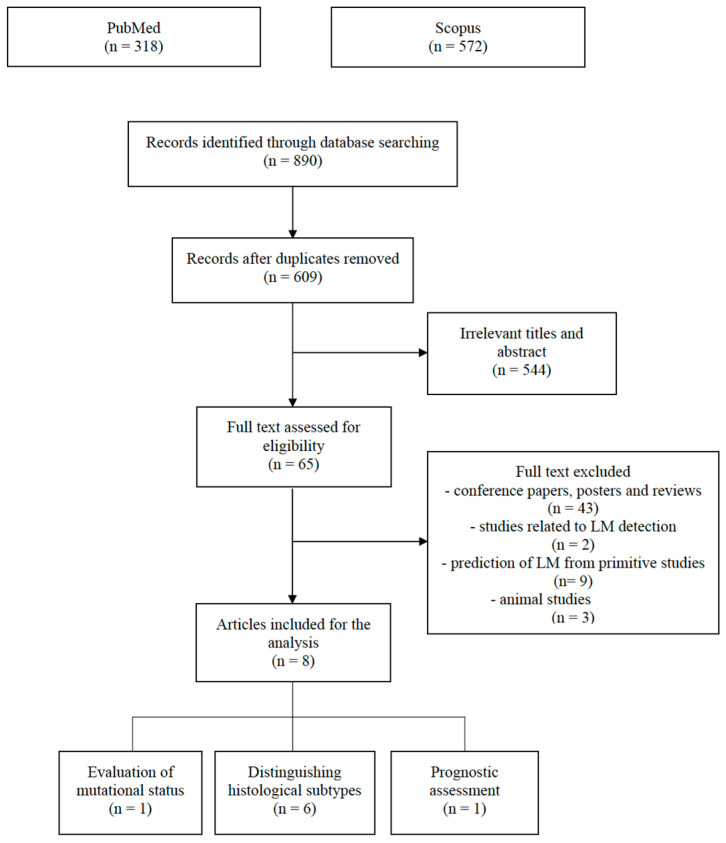
PRISMA flow chart illustrating the inclusion and exclusion criteria for article search.

**Table 1 jpm-13-00225-t001:** Methodology and main findings of the 8 articles selected. ML = Machine Learning.

Authors [Reference #]	Publication Year	Objective	Patient #	Imaging Modality	Segmentation	Feature Selection Method(s)	Validation Method
Angus et al. [71]	2021	To evaluate BRAF mutation in LM from melanoma	103	CT	Semiautomatic (in-house software)	Workflow for Optimal Radiomics Classification (WORC)	100× random-split cross-validation
Hu et al. [72]	2019	To predict LM in colorectal cancer patients with indeterminate pulmonary nodules	194	FDG-PET/CT	Semiautomatic (MIM^®^)	Least Absolute Shrinkage and Selection Operator (LASSO)	Tenfold cross-validation
Kirienko et al. [73]	2018	To differentiate between primary lung tumor and LM lesions and to classify histological subtypes	534	CT	Semiautomatic (PET VCAR^®^)	Linear Discriminant Analysis (LDA)	Direct and backward elimination ×100
Liu et al. [74]	2021	To discriminate benign nodules from LM in patients with colorectal cancer	57	CT	Manual (ITK-SNAP)	Performed by commercial software (AK^®^) after conversion to Co-occurrence of Local Anisotropic Gradient Orientations (CoLIAGe) and combination of Discrete wavelet transform and Local binary pattern (DWT + LBP)	100× cross-validation
Miao et al. [75]	2022	To predict the efficacy of epirubicin combined with ifosfamide in patients with LM from soft tissue sarcoma	51	CT	Manual or semiautomatic (ITK-SNAP)	Random Forest (RF), logistic regression, Support Vector Machine (SVM), naïve Bayesian classification, decision tree classifier, K-nearest neighbor (KNN)	No cross-validation performed on the best ML method (Forest)
Shang et al. [76]	2022	To differentiate LM from different tumor types	78 + 35	CT	Manual	RF, SVM	Tenfold cross-validation
Zhong et al. [77]	2022	To discriminate second primary lung cancers from LM	252	FDG-PET/CT	Semiautomatic (ITK-SNAP)	Minimum Redundancy–Maximum Relevance (mRMR), LASSO / multivariate logistic regression	Calibration curve
Zhou et al. [78]	2021	To differentiate primary lung tumors from LM lesions and to classify histological subtypes	769	CT	Semiautomatic	RF, Distance Correlation (DC), eXtreme gradient boosting (Xgboost), gradient boosting decision tree (GBDT), LASSO	Tenfold cross-validation

**Table 2 jpm-13-00225-t002:** RQS calculation for the 8 articles selected, based on the criteria illustrated by Lambin et al. [70].

Item #	Criteria	Points	Angus et al. [71]	Hu et al. [72]	Kirienko et al. [73]	Liu et al. [74]	Miao et al. [75]	Shang et al. [76]	Zhong et al. [77]	Zhou et al. [78]
1	Image protocol quality—well-documented image protocols (e.g., contrast, slice thickness, energy, etc.) and/or usage of public image protocols allow reproducibility/replicability	+1 (if protocols are well-documented) +1 (if public protocol is used)	1	1	1	1	1	1	1	1
2	Multiple segmentations—possible actions are: segmentation by different physicians/algorithms/software, perturbing segmentations by (random) noise, segmentation at different breathing cycles. Analyze feature robustness to segmentation variabilities	+1	1	1	0	1	1	1	1	0
3	Phantom study on all scanners—detect inter-scanner differences and vendor-dependent features. Analyze feature robustness to these sources of variability	+1	0	0	0	0	0	0	0	0
4	Imaging at multiple time points—collect individuals’ images at additional time points. Analyze feature robustness to temporal variabilities (e.g., organ movement, organ expansion/shrinkage)	+1	0	0	0	0	0	0	0	0
5	Feature reduction or adjustment for multiple testing—decreases the risk of overfitting. Overfitting is inevitable if the number of features exceeds the number of samples. Consider feature robustness when selecting features	−3 (if neither measure is implemented) +3 (if either measure is implemented)	3	3	3	3	3	3	3	3
6	Multivariable analysis with nonradiomic features (e.g., EGFR mutation)—is expected to provide a more holistic model. Permits correlating/inferencing between radiomics and non radiomics features	+1	1	1	0	0	0	0	1	1
7	Detect and discuss biological correlates—demonstration of phenotypic differences (possibly associated with underlying gene–protein expression patterns) deepens understanding of radiomics and biology	+1	0	0	0	0	0	0	0	0
8	Cut-off analyses—determine risk groups by either the median, a previously published cut-off or report a continuous risk variable. Reduces the risk of reporting overly optimistic results	+1	0	0	0	0	0	0	0	0
9	Discrimination statistics—report discrimination statistics (e.g., C-statistic, ROC curve, AUC) and their statistical significance (e.g., *p*-values, confidence intervals). One can also apply resampling method (e.g., bootstrapping, cross-validation)	+1 (if a discrimination statistics and its statistical significance are reported) +1 (if also an resampling method technique is applied)	2	2	2	2	1	2	2	2
10	Calibration statistics—report calibration statistics (e.g., calibration-in-the-large/slope, calibration plots) and their statistical significance (e.g., *p*-values, confidence intervals). One can also apply resampling method (e.g., bootstrapping, cross-validation)	+1 (if a calibration statistics and its statistical significance are reported) +1 (if also an resampling method technique is applied)	0	1	0	0	0	0	1	0
11	Prospective study registered in a trial database—provides the highest level of evidence supporting the clinical validity and usefulness of the radiomics biomarker	+7 (for prospective validation of a radiomics signature in an appropriate trial)	0	0	0	0	0	0	0	0
12	Validation—the validation is performed without retraining and without adaptation of the cut-off value, provides crucial information with regard to credible clinical performance	−5 (if validation is missing) +2 (if validation is based on a dataset from the same institute) +3 (if validation is based on a dataset from another institute) +4 (if validation is based on two datasets from two distinct institutes) +4 (if the study validates a previously published signature) +5 (if validation is based on three or more datasets from distinct institutes) Datasets should be of comparable size and should have at least 10 events per model feature	2	2	2	2	2	3	2	2
13	Comparison to ‘gold standard’—assess the extent to which the model agrees with/is superior to the current ‘gold standard’ method (e.g., TNM-staging for survival prediction). This comparison shows the added value of radiomics	+2	0	0	0	0	0	0	0	0
14	Potential clinical utility—report on the current and potential application of the model in a clinical setting (e.g., decision curve analysis)	+2	0	2	0	0	2	0	2	0
15	Cost-effectiveness analysis—report on the cost-effectiveness of the clinical application (e.g., quality adjusted life years generated)	+1	0	1	0	0	0	0	0	0
16	Open science and data—make code and data publicly available. Open science facilitates knowledge transfer and reproducibility of the study	+1 (if scans are open source) +1 (if region of interest segmentations are open source) +1 (if code is open source) +1 (if radiomics features are calculated on a set of representative ROIs and the calculated features + representative ROIs are open source)	0	0	0	0	0	0	0	0
Sum of scores (%)			9 (25.0%)	13 (36.1%)	8 (22.2%)	14 (38.9%)	9 (25.0%)	10 (27.8%)	10 (27.8%)	10 (27.8%)

## Data Availability

Data can be made publicly available upon reasonable request.

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
