# Peer review of "Radiomics in Lung Metastases: A Systematic Review"

_jpm, 2023, doi:10.3390/jpm13020225_

Round 1

Reviewer 1 Report

Well written, carefully done study 

some suggestions for minor revision

1. How many of the papers are based on machine learning?  I assume that these are mostly ML papers.  Please add to Table  1 the ML method employed, if used

Please expand Table 1 by indicating the validation method (holdout sample, cross validation, etc).  Every ML study claims to do some sort of validation but the rigor of this varies, and that has a big impact on the generalizability of the model.  To avoid leakage and overfitting the validation set must be strictly isolated from the training and data selection algorithm, which is often not the case. Other requirements for valid ML are in  Volovici et al https://doi.org/10.1038/s41591-022-01961-6

2. Please give more information how the RQS was determined (Table 1). Was this based on votes of multiple scorers?  It would be best to include the questions (cite 70) together with scores for each question for the 8 studies. Which criteria were most commonly not satisfied?

3. The AUC for the algorithms were cited to 3 significant digits. How reliable is this? Most "canned" ML programs calculate an AUC, but commonly use only 3 points - a single sensitivity/specificity value and endpoints (0,0) and (1,1). That would hardly justify citing AUC to 3 significant digits.  I wonder if the differences in AUC in the different papers is meaningful given this crude assessment?

Reviewer 2 Report

The authors need to include the clinical implications of the radiomics and how it can improve the treatment modalities as well as elaborate the future directions a bit.

I would recommend acceptance of this manuscript after inclusion of these changes.
